# Investigating Lignin-Derived Monomers and Oligomers in Low-Molecular-Weight Fractions Separated from Depolymerized Black Liquor Retentate by Membrane Filtration

**DOI:** 10.3390/molecules26102887

**Published:** 2021-05-13

**Authors:** Kena Li, Jens Prothmann, Margareta Sandahl, Sara Blomberg, Charlotta Turner, Christian Hulteberg

**Affiliations:** 1Department of Chemical Engineering, Lund University, SE-21100 Lund, Sweden; kena.li@chemeng.lth.se (K.L.); sara.blomberg@chemeng.lth.se (S.B.); 2Centre for Analysis and Synthesis, Department of Chemistry, Lund University, SE-22100 Lund, Sweden; Jens.prothmann@chem.lu.se (J.P.); margareta.sandahl@chem.lu.se (M.S.); Charlotta.Turner@chem.lu.se (C.T.)

**Keywords:** black liquor retentate, depolymerization, identification, monomer, oligomers

## Abstract

Base-catalyzed depolymerization of black liquor retentate (BLR) from the kraft pulping process, followed by ultrafiltration, has been suggested as a means of obtaining low-molecular-weight (LMW) compounds. The chemical complexity of BLR, which consists of a mixture of softwood and hardwood lignin that has undergone several kinds of treatment, leads to a complex mixture of LMW compounds, making the separation of components for the formation of value-added chemicals more difficult. Identifying the phenolic compounds in the LMW fractions obtained under different depolymerization conditions is essential for the upgrading process. In this study, a state-of-the-art nontargeted analysis method using ultra-high-performance supercritical fluid chromatography coupled to high-resolution multiple-stage tandem mass spectrometry (UHPSFC/HRMS^n^) combined with a Kendrick mass defect-based classification model was applied to analyze the monomers and oligomers in the LMW fractions separated from BLR samples depolymerized at 170–210 °C. The most common phenolic compound types were dimers, followed by monomers. A second round of depolymerization yielded low amounts of monomers and dimers, while a high number of trimers were formed, thought to be the result of repolymerization.

## 1. Introduction

It is widely accepted that we must find a renewable resource to replace petroleum-based raw materials for the production of energy and chemicals. Lignin is one of the three main components of biomass and is the most abundant natural source of aromatic polymers and has attracted attention in this respect [1,2]. The most abundant source of lignin is currently from kraft pulping, which is used in almost 90% of the world’s pulp production [3]. Today, the kraft lignin generated by the kraft process is usually burned for internal energy supply, but modern pulp mills allow the extraction of a fraction of this lignin without disturbing the operation of the mill [4,5]. 

The first and most important step in lignin valorization is to break down the lignin macromolecules into smaller compounds. The complexity of the lignin molecule itself and the condensation reactions during the kraft process increase the complexity and recalcitrance of kraft lignin, making depolymerization more difficult. Numerous studies have been carried out on the depolymerization of kraft lignin during recent decades, including homogeneous base-catalyzed depolymerization (BCD) using NaOH as the catalyst [6,7,8,9,10], acid-catalyzed depolymerization [11,12], ionic liquid-catalyzed depolymerization [13,14,15,16], oxidative depolymerization using oxidants or an oxidizing protocol [17,18,19,20], and heterogeneously catalyzed depolymerization using supported transition-metal catalysts (e.g., Ni, Mo, etc.) [21,22,23]. BCD could be a promising method for the depolymerization of the kraft lignin extracted from black liquor using membrane filtration, which is dissolved in the alkaline solution. Performing BCD in a continuous flow system could suppress repolymerization of the depolymerized components [24]. In addition to depolymerization, effective techniques are required to separate the lignin degradation products from the reaction solution. Acid can be used to lower the pH of base-catalyzed depolymerized kraft lignin and precipitate the high-molecular-weight compounds. However, a high amount of acid would be needed in the precipitation process, which would consume the NaOH in the liquor. Extensive research has been carried out on membrane filtration for the extraction of lignin from black liquor [4,25], and this could also be an attractive approach for extracting low-molecular-weight (LMW) compounds from depolymerized kraft lignin. 

In a previous study, we developed a process combining base-catalyzed continuous-flow depolymerization of kraft lignin and membrane separation to obtain LMW compounds. Using this approach, over 70% of the high-molecular-weight polymers were removed, providing a permeate rich in LMW compounds [26]. Before further upgrading the LMW permeate, it is necessary to identify the aromatic compounds present. Gas chromatography, high-performance liquid chromatography (HPLC), and ultra-high-performance supercritical fluid chromatography (UHPSFC) coupled with mass spectrometry (MS) have been developed to analyze lignin-derived monomers [27,28,29]. In contrast, because of the lack of commercially available reference standards, the identification of oligomers is more challenging. Self-synthesized oligomer standards were often used to study lignin oligomer characteristic fragmentation patterns [30]. Studies using MS fragmentation patterns for the identification of oligomers have been reported [31,32]. We have also previously developed a nontargeted analysis method using UHPSFC-high-resolution mass spectrometry (HRMS) combined with Kendrick mass defect-based principal component analysis-quadratic discriminant analysis (KMD-PCA-QDA) classification models and used it for the analysis of both monomers and oligomers in different lignin samples [33]. This nontargeted method would be useful in evaluating different BLR permeates from membrane filtration.

The aim of this study was therefore to use the nontargeted UHPSFC-HRMS^n^ method developed in our previous study to analyze the LMW compound-rich permeates obtained from membrane filtration of BLR depolymerized at different temperatures using BCD. As a classification model, KMD-PCA-QDA, was used to identify the monomers and oligomers. The effect of BCD at different temperatures is compared, and the multiple depolymerization and membrane filtration on the lignin monomers and dimers production is discussed.

## 2. Results and Discussion

### 2.1. Depolymerization and Ultrafiltration of BLR: Initial Analysis of Lignin Monomers

Valorization of the BLR sample, a kraft lignin from a ‘real’ paper and pulp process, increases the value of the mill operation. The combination of depolymerization and membrane filtration was investigated previously in order to obtain lignin-derived LMW compounds [26]. After ultrafiltration of the BLR depolymerized at 170, 190, and 210 °C, three permeates (BDM170, BDM190, and BDM210) were obtained. The retentate from the ultrafiltration of the depolymerized BLR at 190 °C was depolymerized the second time, and the permeate (2BDM190) was obtained after the second ultrafiltration of the depolymerized retentate. These permeates contained a number of monomers and oligomers, which, once identified, will have the potential to be converted into valuable chemicals or fuels through bioconversion, or further catalytic conversion. 

In order to evaluate the abundance of lignin monomers in the different permeates, UHPLC with UV detection was used for analysis. Figure 1 shows the concentration of the two main monomers, guaiacol and vanillin, in the four permeate samples, together with the chromatograms from UHPLC. It can be seen that higher amounts of monomers were obtained at higher depolymerization temperatures. The guaiacol and vanillin concentrations in sample 2BDM190 were relatively low, demonstrating that the efficiency of the second round of depolymerization was lower than the first. 

### 2.2. Nontargeted Identification of LMW Compounds from Depolymerized BLR

In order to enable the identification of other compounds, especially lignin dimers and trimers, a recently developed non-targeted analysis method for the identification of the aromatic compounds in technical lignin samples was used [33]. This method provides important information about structural changes during depolymerization, and a deeper understanding of how to tailor downstream conversion. The base peak ion-chromatograms of BDM190 and 2BDM190 are shown in Figure 2. It can be seen that UHPSFC gives the average peak width of approximately 0.2–0.3 min, while UHPLC (Figure 1) show peaks of up to 0.5 min in width. This can be explained by the lower viscosity of the mobile phase in UHPSFC, giving more efficient chromatography. 

Table 1 gives the number of validated *m*/*z* values being either lignin monomers, dimers, or trimers, the number of *m*/*z* values with more than one retention time, and the number of aromatic compounds identified as monomers, dimers, or trimers in the four depolymerized BLR samples. Details including the detected *m*/*z*, obtained chemical formula, classified class, obtained mass difference, obtained ^13^C ratio, ring double bond equivalent (RDB), retention time, obtained MS^n^ spectra, suggested compounds, and obtained identification confidence level of all identified monomers, dimers, and trimers can be found in Appendix A Appendix A for the BDM170 sample [33], in Appendix A Appendix A for the BDM190 sample, in Appendix A Appendix A for the BDM210 sample, and in Appendix A Appendix A for the 2BDM190 sample. The total number of identified and validated lignin monomers, dimers, and trimers increases with increasing reaction temperature during the depolymerization process, from 44 *m*/*z* values at a temperature of 170 °C to 65 and 79 *m*/*z* values at 190 °C and 210 °C, respectively. These results show that under the experimental conditions used, a higher temperature resulted in greater depolymerization, which is in accordance with the molecular weight distribution results reported in our previous study [26]. Many of the identified and validated *m*/*z* values in the depolymerized kraft lignin samples showed more than one retention time in their corresponding extracted-ion chromatogram (EIC), meaning that structural isomers are present. Both structural diversity in the lignin and different depolymerization reactions can cause structural isomers of the lignin degradation products [34,35]. When including the structural isomers, 77 lignin-derived phenolic compounds were identified in the BDM170 sample, 135 in BDM190, and 186 in the BDM210 sample. The number of *m*/*z* values with more than one retention time in the EIC also increased with increasing depolymerization temperature. 

Dimers were the dominant class of compounds identified, followed by monomers. No lignin trimers were identified in BDM170 or BDM190, and only one lignin trimer was identified in the BDM210 sample. The UHPSFC-HRMS^n^ method used has been proved to enable analysis of lignin oligomers up to tetramers in a previous study. For example, 38 trimers and 5 tetramers were identified from the LignoBoost kraft lignin [33]. The trimers and tetramers may be lost during the acid precipitation process as they do not show up on the analysis. The percentage of monomers relative to dimers increased with increasing depolymerization temperature. In the BDM170 sample, 22% of the identified phenolic compounds were monomers; in the BDM190 sample, 32%, and in the BDM210 sample, 52%. The increase in the number of phenolic compounds identified, the increase in structural isomers, and the increase in the relative amount of lignin monomers can be attributed to greater reactivity in the depolymerization process at higher temperatures, leading to more chemical transformation reactions. At the lowest temperature, 170 °C, depolymerization is incomplete, as confirmed by the 2D HSQC NMR results in our previous study, showing that β-O-4, β-β, and β-5 bonds exist at this temperature [36]. When the temperature was increased to 190 °C, double the amounts of both monomers and dimers were generated, indicating the cleavage of additional C-O and C-C bonds at this higher temperature. Increasing the temperature further, to 210 °C, can lead to a higher degree of depolymerization of large molecules to dimers, while a few dimers could be degraded to monomers. 

To better visualize the chemical differences between the different BLR samples, a van Krevelen plot was constructed (Figure 3). It can be seen that at the highest depolymerization temperature of 210 °C, there are more chemical formulas with a higher carbon content than at the other two depolymerization temperatures. Chemical formulas with relatively high carbon contents, such as C_13_H_8_O_2_ with 57% carbon, or C_13_H_8_O_4_ with 52% carbon content, were only identified in the BDM210 sample. The presence of such compounds in the BDM210 sample may be due to more decarboxylation and dehydration reactions at higher temperatures. Figure 3 also shows an example of a decarboxylation reaction of C_14_H_10_O_5,_ leading to C_13_H_10_O_3_, followed by dehydration leading to C_13_H_8_O_2_. Compounds with the chemical formula C_14_H_10_O_5_ were found in the BDM170 and BDM210 samples, while C_13_H_10_O_3_ was found in BDM190 and BDM210. A compound with the chemical formula C_13_H_8_O_2_, which may be a product of decarboxylation and dehydration reactions of C_14_H_10_O_5_, was only found in the BDM210 sample. 

The lignin-derived aromatic compounds identified in the 2BDM190 sample show that a few aromatic compounds were produced in the second depolymerization step. Comparing the phenolic compounds identified in the BDM190 sample with those in the 2BDM190 sample shows that after separating the LMW compounds from the depolymerized products, the heavy lignin fractions can be further depolymerized to some extent (see Table 1). However, the depolymerization efficiency was considerably lower than in the first depolymerization round. Only nine different aromatic monomers were identified. The most probable reason for this is the mild depolymerization conditions. Due to the condensed lignin structure resulting from the kraft process, the structures that can be further degraded under this mild condition are limited. Repeating the depolymerization treatment does not lead to the degradation of large lignin molecules. Furthermore, additional heat treatment may lead to the repolymerization of monomers into dimers or trimers. This explanation is further reinforced by the lower number of monomers found in 2BDM190 than in BDM190, however containing more trimers. 

A van Krevelen plot comparing the *m*/*z* values identified in the BDM190, and the 2BDM190 samples are shown in Figure 4. It can be seen here that the identified compounds are projected mainly within the same area, illustrating that similar compounds can be generated at the same depolymerization temperature. A small number of new compounds were only identified in the 2BDM190 sample. However, some compounds were only identified in either the BDM190 or 2BDM190, illustrating that different types of lignin dimers are produced after the second round of depolymerization.

### 2.3. Characterization of the Monomers and Oligomers

The tentative structures of the lignin dimers in each sample were proposed based on MS^n^ experiments, with results shown in Table 2. The proposed structures either show a β-O-5 linkage with variation in type and number of functional groups, a direct connection of the benzene rings by a 5-5 linkage or linkage of the benzene rings by a 3-O-5 ether linkage. No clear difference in terms of dimer structures connected to the reaction temperature used was found. The identified tentative structures in the four sample were similar. Among the 13 tentative structures, 7 of those were found in all of the permeate samples. As can be seen from Table 2, the tentative structures of dimers no. 3, 7, and 12 that were present in BDM190 cannot be detected in the 2BDM190 sample. This result shows that the dimers obtained from the first round of depolymerization were basically filtered by the membrane. Although under the same depolymerization condition, the dimers from the second round depolymerization are different. Furthermore, dimer no. 13 was only detected in 2BDM190. As reported in our previous research, still some monomers can be identified in the retentate sample. This new dimer in 2BDM190 is probably produced due to the repolymerization of the monomers in the retentate during the second round of depolymerization. 

## 3. Materials and Methods

### 3.1. Lignin Samples Preparation

A black liquor retentate (BLR), i.e., a mixture of softwood and hardwood lignin, was obtained as a thick liquid from a Swedish membrane filtration pilot plant and was used as raw material. The source of the material, the characterization methods, and the results related to BLR have been reported in our previous research [26]. It contains 32.9% of total solid, 22.4% of total lignin, and 6.5% of ash. 

The sample production has been described in detail previously [26]. Briefly, the feed prior to the experimental run was prepared by diluting the BLR 5 times with 2% NaOH solution. The total solids, total lignin, ash, and hemicellulose content of the feed were 84.1, 47.2, 25.9, and 5.52 g/L, respectively. The BLR were subjected to BCD, followed by ultrafiltration to obtain the LMW compounds. A schematic flow diagram of the sample production and preparation process is shown in Figure 5. BCD was carried out in a continuous flow reactor at 170, 190, or 210 °C with a residence time of 2 min. The depolymerized samples are denoted BD170, BD190, and BD210. An Alfa Laval membrane, GR95PP, with a molecular weight cut-off of 2 kDa was used to separate the LMW compounds from the depolymerized BLR. After ultrafiltration, the permeates, denoted BDM170, BDM190, and BDM210, were collected for further extraction and analysis. The retentate obtained from BD190 after ultrafiltration was diluted 5 times with NaOH solution (2%), and was used as the feed for the second depolymerization and ultrafiltration process, under the same conditions as the first. The permeate, 2BDM190, was then collected for further treatment and analysis. 

Four permeate samples (BDM170, BDM190, BDM210, and 2BDM190) were neutralized with 5 M HCl to pH 6–7 and centrifuged (Figure 5). For guaiacol and vanillin analysis, the supernatants were collected and filtered using 0.2 µm CA syringe filters. For UHPSFC/HRMS^n^ analysis, 2 × 1 mL ethyl acetate was used for liquid–liquid extraction of the phenolic compounds from 1 mL of each supernatant. All samples were centrifuged for 10 min at 20 °C and 14,000 rpm before being injected into the UHPSFC/HRMS^n^ system for analysis.

### 3.2. Chemicals

Acetone (HPLC grade) and acetonitrile (HPLC-MS grade) were obtained from VWR (Radnor, PA, USA). Ammonium formate (LC-MS) grade, NaOH, and HCl (37%) were purchased from Sigma-Aldrich (St. Louis, MO, USA). Methanol and ethyl acetate (both LC-MS grade) were purchased from J. T. Baker (Philipsburg, NJ, USA) and Merck (Darmstadt, Germany), respectively. Carbon dioxide (scientific grade, 5.2) was obtained from Linde (Munich, Germany). 

### 3.3. Equipment

The UHPLC experiments were performed on a Waters H-Class UHPLC system using a modified method described by Krithika et al. [37]. The UHPLC system was equipped with a Waters BEH C18 column (100 × 2.1 mm, 1.7 µm) and run at 50 °C. Detection was performed using a PDA detector at 280 nm, with the baseline-corrected by 350–400 nm. The injection volume varied between 3 and 5 µL, and the flow rate was 0.6 mL/min. The mobile phase was composed of: (A) Milli-Q water with 0.5% acetic acid or 10 mM formic acid, and (B) acetonitrile with 0.5% acetic acid or 10 mM formic acid.

All UHPSFC/HRMS experiments were performed using a Waters Ultra Performance Convergence Chromatography System (Waters Milford, MA, USA) connected via a flow splitter (ACQUITY UPC^2^ splitter, Waters) to an LTQ Orbitrap Velos Pro mass spectrometer (Thermo Scientific, Waltham, MA, USA), which was equipped with a heated electrospray ionization source (HESI, Thermo Scientific). Chromatographic separation of the samples was performed using a Torus DIOL column (3 mm × 100 mm, 1.7 µm, Waters) protected with a Torus DIOL VanGuard pre-column (2.1 mm × 5 mm, 1.7 µm, Waters). The samples were centrifuged in a 5424R Eppendorf centrifuge (Eppendorf, Hamburg, Germany).

### 3.4. UHPSFC/HRMS and UHPSFC/HRMS^n^ Methods

The UHPSFC/HRMS and UHPSFC/HRMS^n^ methods were based on a non-targeted analysis method tailored for aromatic compounds, which was developed previously by our group [33]. In brief, a DIOL column was used for sample separation, with an injection volume of 1.5 µL, a flow rate of 2 mL/min, a column temperature of 50 °C and a back pressure of 130 bar. Gradient elution was applied starting with 0 vol.% B and ramped up to 8.5 vol.% B until 2.5 min, followed by ramping up to 25 vol.% until 5.5 min. B was then maintained at 25 vol.% until 7.5 min, and then decreased to the starting conditions until 8.0 min. The starting conditions were then maintained for 2 min to ensure column equilibration. The mobile phase consisted of CO_2_ (A) and methanol (B). The makeup solvent was 10 mmol/L ammonium formate in methanol, and a makeup solvent flow rate of 0.6 mL/min was used. ESI was used in negative mode with a source temperature of 275 °C, a sheath gas flow rate of 80 arbitrary units (AU), an auxiliary gas flow rate of 20 AU, a capillary voltage of 3.0 kV and a capillary temperature of 275 °C. The mass range was screened from *m*/*z* 80 to 1500. A full MS scan was performed for each sample with the MS resolution set to 60,000. Furthermore, five data-dependent neutral-loss MS^3^ experiments were performed for each sample. The screened neutral losses were a methyl radical (CH_3_), water (H_2_O), carbon dioxide (CO_2_), formic acid (CH_2_O_2_), and formaldehyde plus water (CH_2_O + H_2_O). The MS resolution was set to 30,000 at all MS^n^ stages.

### 3.5. MS Data Evaluation and Identification of Phenolic Compounds

A previously developed Kendrick mass defect (KMD)-based principal component analysis combined with quadratic discriminant analysis (PCA-QDA) classification model for lignin-derived monomers, dimers, trimers, and tetramers was used for MS data evaluation and the identification of phenolic compounds in the depolymerized lignin samples [33]. The classified *m*/*z* values were validated using ring double-bond equivalents, mass differences between theoretical and detected exact masses, and ratios between theoretical and detected ^13^C peak intensities. A detailed description of the data evaluation has been given by Prothmann et al. [33]. In-source fragmentation, MS^2^ and MS^3^ fragmentation patterns were used for structural elucidation of classified and validated phenolic compounds.

An identification confidence level was assigned to each identified compound according to the identification confidence level system developed by Schymanski et al. [38]. The best identification confidence level is level 1 (confirmed structure), which is assigned when a compound of interest can be identified by comparison with a reference standard. Level 2 (probable structure) is achieved when a probable structure of a compound of interest can be proposed based on MS and MS^n^ data as well as a library spectrum match or diagnostic evidence such as the observation of characteristic neutral losses. At level 3 (tentative candidate), parts of the structure or the compound class are known based on MS data MS^n^ data and other experimental data. At level 4 (unequivocal molecular formula), the chemical formula based on exact mass measurements and the isotopic pattern of the compound of interest are known, whereas at level 5 (exact mass) only the exact mass is known. 

### 3.6. Software

MassLynx 4.1 software (Waters) was used to operate the SFC system and Xcalibur 2.2 (Thermo Fisher Scientific, Waltham, MA, USA) to operate the MS system. Data evaluation was performed using MZmine2 software (open-source) and Xcalibur 2.2. A classification toolbox (Milano Chemometrics and QSAR Research Group, University of Milan, Milan, Italy) for MATLAB (MathWorks, Natick, MA, USA) was used to create and apply the KMD-PCA-QDA classification models. 

## 4. Conclusions

The LMW fraction separated using membrane filtration of BLR samples depolymerized at various temperatures has been analyzed using the newly developed non-targeted UHPSFC/HRMS^n^ method. The results showed that dimers are the most common phenolic compound type. A greater degree of depolymerization was obtained at higher temperatures, leading to the production of more monomers and dimers. Compounds with a higher carbon content were formed after depolymerization at the highest temperature, probably due to decarboxylation and dehydration reactions under more severe condition. Repeating the depolymerization of the heavy fraction remaining after membrane filtration of BLR depolymerized at 190 °C produced lower numbers of monomers and dimers than the first polymerization round, but a high number of trimers were identified. This is most likely due to repolymerization during the repeated treatment. The identification of lignin monomers and oligomers is meaningful for both understanding of kraft lignin conversion process and the future utilization in high-value applications. 

## Figures and Tables

**Figure 1 molecules-26-02887-f001:**
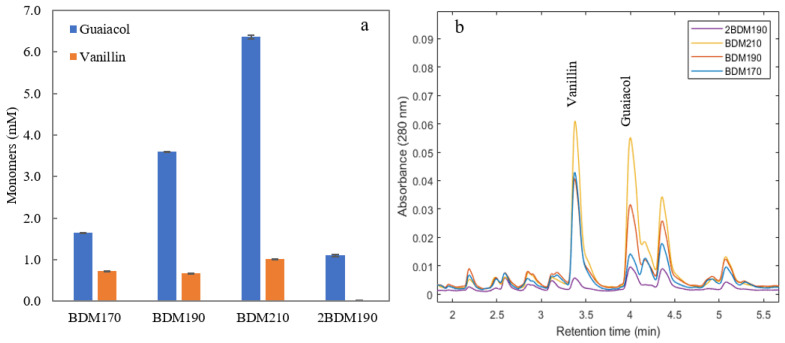
(**a**) Guaiacol and vanillin concentrations in the four permeate samples after membrane filtration of BLR depolymerized at three temperatures (BDM170, BDM190, BDM210, and 2BDM190). (**b**) The chromatograms obtained from UHPLC of the four samples. (Depolymerization was performed at temperatures of 170, 190, and 210 °C with 2 min residence time. The 2BDM190 sample is the permeate after ultrafiltration of the retentate from the first ultrafiltration of the BLR depolymerized at 190 °C).

**Figure 2 molecules-26-02887-f002:**
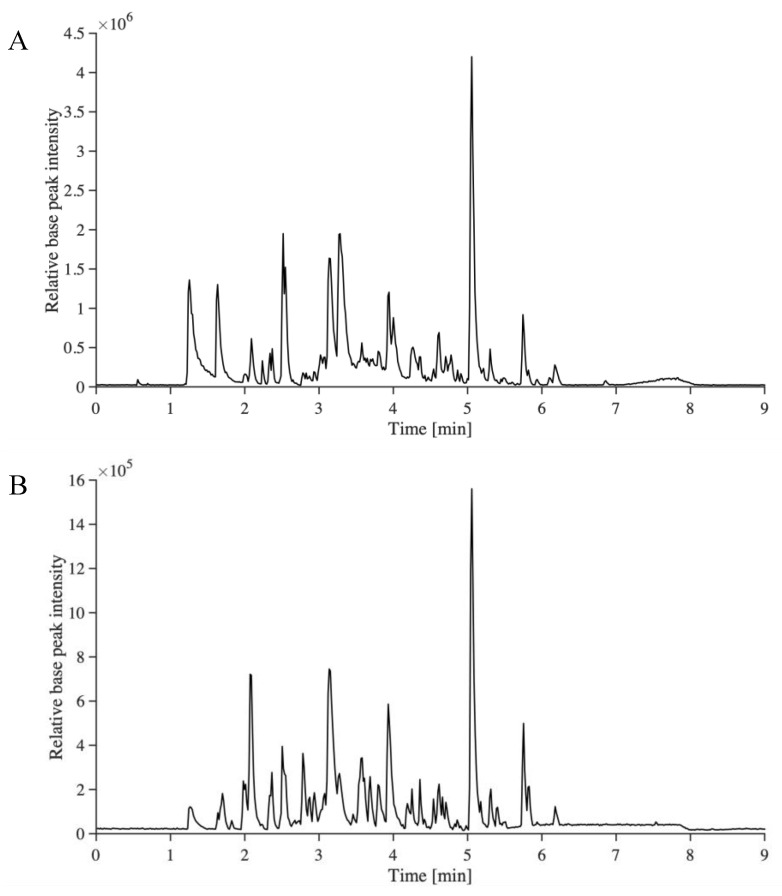
The base peak ion-chromatograms of BDM190 (**A**) and 2BDM190 (**B**) using the developed UHPSFC/HRMS^n^ method.

**Figure 3 molecules-26-02887-f003:**
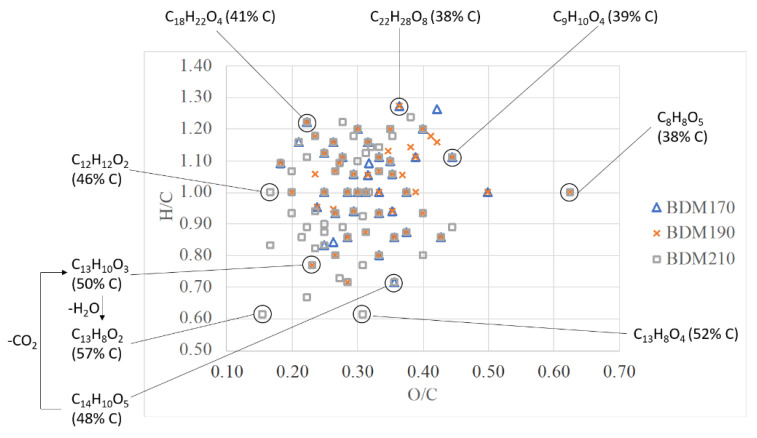
A van Krevelen plot including the *m*/*z* values of lignin-derived phenolic compounds identified in the BDM170, BDM190, and BDM210 samples with the chemical formula and the relative carbon content of selected identified *m*/*z* values.

**Figure 4 molecules-26-02887-f004:**
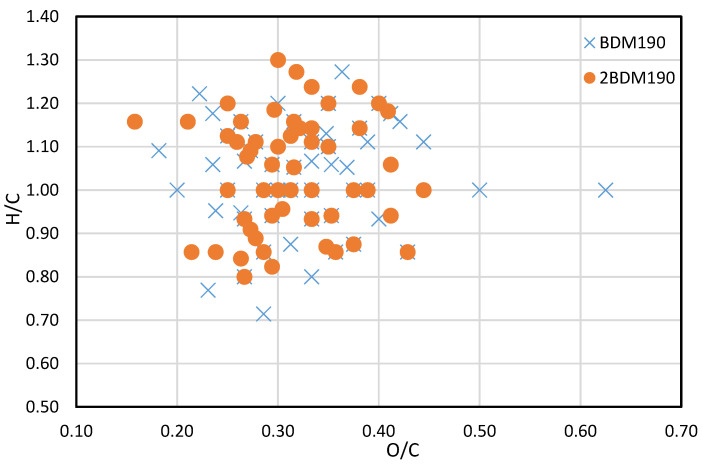
Van Krevelen plot of BDM190 and 2BDM190 samples.

**Figure 5 molecules-26-02887-f005:**
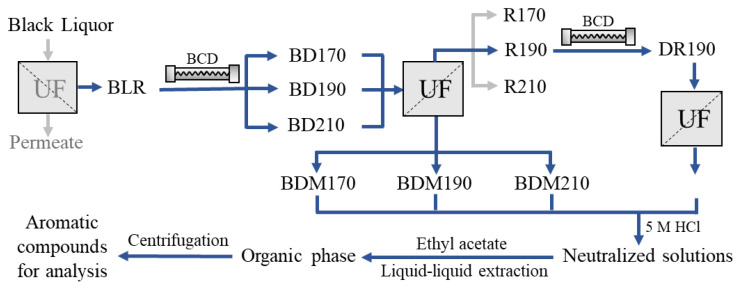
Schematic diagram of BLR sample production and the separation and preparation of phenolic compounds for analysis.

**Table 1 molecules-26-02887-t001:** The number of validated *m*/*z* values identified as lignin monomers, dimers, or trimers, the number of *m*/*z* values with more than one retention time, and the number of aromatic compounds identified as monomers, dimers, or trimers in the depolymerized BLR samples.

Sample	BDM170 *	BDM190	BDM210	2BDM190
Identified and validated *m*/*z* values	44	65	79	58
Monomers	10	17	27	6
Dimers	34	48	51	48
Trimers	0	0	1	4
*m*/*z* values with more than one retention time	20	63	79	58
Identified phenolic compounds	77	135	186	101
Monomers	14	33	63	9
Dimers	63	102	122	84
Trimers	0	0	1	8

* Previously published in [33].

**Table 2 molecules-26-02887-t002:** Overview of obtained tentative structures of dimers detected in the BDM170, BDM190, BDM210, and 2BDM190 samples (+:present; -: not present).

No.	Determined Chemical Formula	Proposed Structure	Detected [M-H]^−^	RT min	BDM170	BDM190	BDM210	2BDM190
1	C_14_H_11_O_5_	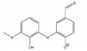	259.0605	3.57	+	+	+	+
2	C_15_H_13_O_5_	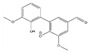	273.0759	3.58	+	+	+	+
3	C_16_H_15_O_5_	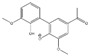	287.0915	2.94	+	+	+	-
4	C_16_H_13_O_6_	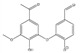	301.0706	3.93	+	+	+	+
5	C_17_H_17_O_5_	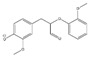	301.1071	2.99	+	+	+	+
6	C_17_H_17_O_6_	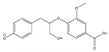	317.1022	4.50	+	-	+	-
7	C_17_H_17_O_6_	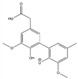	317.1021	4.88	-	+	+	-
8	C_18_H_17_O_6_	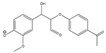	329.1017	3.80	+	+	+	+
9	C_18_H_19_O_6_	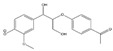	331.1175	4.69	+	+	+	+
10	C_18_H_19_O_7_	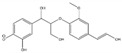	347.1125	4.10	+	+	-	+
11	C_19_H_21_O_6_	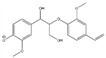	345.1333	4.62	+	+	+	+
12	C_20_H_23_O_6_	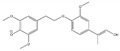	359.1491	4.47	+	+	-	-
13	C_20_H_25_O_6_	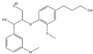	361.1617	5.06	-	-	-	+

## Data Availability

The data presented in this study are available on request from the corresponding author.

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
