# Peer review of "Investigating Lignin-Derived Monomers and Oligomers in Low-Molecular-Weight Fractions Separated from Depolymerized Black Liquor Retentate by Membrane Filtration"

_molecules, 2021, doi:10.3390/molecules26102887_

Round 1
Reviewer 1 Report
Comments are in the attached file

Reviewer 2 Report
The reviewed manuscript deals with lignin depolymerization, the topic of numerous papers. However, this specific study has a distinct novelty and originality. Namely, it is based on thorough GC analysis of the products, up to and including phenolic trimers. Furthermore, it reports the results of the second depolymerization run on the products obtained by a prior lignin depolymerization. The original method of analysis developed by the authors is not without limitations (some analytes can be lost), but it certainly adds some new and valuable information. The study is conducted thoroughly, the methods used are sound, the results’ interpretation is logical, the presentation is solid and succinct. The manuscript is written in good idiomatic English. I thus recommend publication with a minor revision targeting some additional editing. I am attaching a scanned file with some suggested minor editing. I also have one recommendation (non-binding): I would replace the sample numbers with their description, i.e., “200 °C run”, “second 210 °C run,” etc. This way, the reader would not have to look back into the sample description every time s/he encounters a sample number.
